# Monte-Carlo Tree Search by Best Arm Identification

**Emilie Kaufmann**
CNRS & Univ. Lille, UMR 9189 (CRIStAL), Inria SequeL
Lille, France
emilie.kaufmann@univ-lille1.fr

**Wouter M. Koolen**
Centrum Wiskunde & Informatica,
Science Park 123, 1098 XG Amsterdam, The Netherlands
wmkoolen@cwi.nl

## Abstract

Recent advances in bandit tools and techniques for sequential learning are steadily enabling new applications and are promising the resolution of a range of challenging related problems. We study the game tree search problem, where the goal is to quickly identify the optimal move in a given game tree by sequentially sampling its stochastic payoffs. We develop new algorithms for trees of arbitrary depth, that operate by summarizing all deeper levels of the tree into confidence intervals at depth one, and applying a best arm identification procedure at the root. We prove new sample complexity guarantees with a refined dependence on the problem instance. We show experimentally that our algorithms outperform existing elimination-based algorithms and match previous special-purpose methods for depth-two trees.

## 1   Introduction

We consider two-player zero-sum turn-based interactions, in which the sequence of possible successive moves is represented by a maximin game tree $\mathcal{T}$. This tree models the possible actions sequences by a collection of MAX nodes, that correspond to states in the game in which player A should take action, MIN nodes, for states in the game in which player B should take action, and leaves which specify the payoff for player A. The goal is to determine the best action at the root for player A. For deterministic payoffs this search problem is primarily algorithmic, with several powerful pruning strategies available [20]. We look at problems with stochastic payoffs, which in addition present a major statistical challenge.

Sequential identification questions in game trees with stochastic payoffs arise naturally as robust versions of bandit problems. They are also a core component of Monte Carlo tree search (MCTS) approaches for solving intractably large deterministic tree search problems, where an entire sub-tree is represented by a stochastic leaf in which randomized play-out and/or evaluations are performed [4]. A play-out consists in finishing the game with some simple, typically random, policy and observing the outcome for player A.

For example, MCTS is used within the AlphaGo system [21], and the evaluation of a leaf position combines supervised learning and (smart) play-outs. While MCTS algorithms for Go have now reached expert human level, such algorithms remain very costly, in that many (expensive) leaf evaluations or play-outs are necessary to output the next action to be taken by the player. In this paper, we focus on the *sample complexity* of Monte-Carlo Tree Search methods, about which very little is known. For this purpose, we work under a simplified model for MCTS already studied by [22], and that generalizes the depth-two framework of [10].

## 1.1 A simple model for Monte-Carlo Tree Search

We start by fixing a game tree $\mathcal{T}$, in which the root is a MAX node. Letting $\mathcal{L}$ be the set of leaves of this tree, for each $\ell \in \mathcal{L}$ we introduce a stochastic oracle $\mathcal{O}_\ell$ that represents the leaf evaluation or play-out performed when this leaf is reached by an MCTS algorithm. In this model, we do not try to optimize the evaluation or play-out strategy, but we rather assume that the oracle $\mathcal{O}_\ell$ produces i.i.d. samples from an unknown distribution whose mean $\mu_\ell$ is the value of the position $\ell$. To ease the presentation, we focus on binary oracles (indicating the win or loss of a play-out), in which the oracle $\mathcal{O}_\ell$ is a Bernoulli distribution with unknown mean $\mu_\ell$ (the probability of player A winning the game in the corresponding state). Our algorithms can be used without modification in case the oracle is a distribution bounded in $[0,1]$.

For each node $s$ in the tree, we denote by $\mathcal{C}(s)$ the set of its children and by $\mathcal{P}(s)$ its parent. The root is denoted by $s_0$. The *value* (for player A) of any node $s$ is recursively defined by $V_\ell = \mu_\ell$ if $\ell \in \mathcal{L}$ and

$$V_s = \begin{cases} \max_{c \in \mathcal{C}(s)} V_c & \text{if s is a MAX node,} \\ \min_{c \in \mathcal{C}(s)} V_c & \text{if s is a MIN node.} \end{cases}$$

The best move is the action at the root with highest value,

$$s^* = \operatorname*{argmax}_{s \in \mathcal{C}(s_0)} V_s.$$

To identify $s^*$ (or an $\epsilon$-close move), an MCTS algorithm sequentially selects paths in the game tree and calls the corresponding leaf oracle. At round $t$, a leaf $L_t \in \mathcal{L}$ is chosen by this adaptive *sampling rule*, after which a sample $X_t \sim \mathcal{O}_{L_t}$ is collected. We consider here the same PAC learning framework as [22, 10], in which the strategy also requires a *stopping rule*, after which leaves are no longer evaluated, and a *recommendation rule* that outputs upon stopping a guess $\hat{s}_\tau \in \mathcal{C}(s_0)$ for the best move of player A.

Given a risk level $\delta$ and some accuracy parameter $\epsilon \geq 0$ our goal is have a recommendation $\hat{s}_\tau \in \mathcal{C}(s_0)$ whose value is within $\epsilon$ of the value of the best move, with probability larger than $1 - \delta$, that is

$$\mathbb{P}\left(V(s_0) - V(\hat{s}_\tau) \leq \epsilon\right) \geq 1 - \delta.$$

An algorithm satisfying this property is called $(\epsilon, \delta)$-correct. The main challenge is *to design $(\epsilon, \delta)$-correct algorithms that use as few leaf evaluations $\tau$ as possible.*

**Related work** The model we introduce for Monte-Carlo Tree Search is very reminiscent of a stochastic bandit model. In those, an agent repeatedly selects one out of several probability distributions, called arms, and draws a sample from the chosen distribution. Bandits models have been studied since the 1930s [23], mostly with a focus on *regret minimization*, where the agent aims to maximize the sum of the samples collected, which are viewed as rewards [18]. In the context of MCTS, a sample corresponds to a win or a loss in one play-out, and maximizing the number of successful play-outs (that correspond to simulated games) may be at odds with identifying quickly the next best action to take at the root. In that, our best action identification problem is closer to a so-called *Best Arm Identification* (BAI) problem.

The goal in the standard BAI problem is to find quickly and accurately the arm with highest mean. The BAI problem in the fixed-confidence setting [7] is the special case of our simple model for a tree of depth one. For deeper trees, rather than finding the best arm (i.e. leaf), we are interested in finding the best action at the root. As the best root action is a function of the means of all leaves, this is a more structured problem.

Bandit algorithms, and more recently BAI algorithms have been successfully adapted to tree search. Building on the UCB algorithm [2], a regret minimizing algorithm, variants of the UCT algorithm [17] have been used for MCTS in growing trees, leading to successful AIs for games. However, there are only very weak theoretical guarantees for UCT. Moreover, observing that maximizing the number of successful play-outs is not the target, recent work rather tried to leverage tools from the BAI literature. In [19, 6] Sequential Halving [14] is used for exploring game trees. The latter algorithm is a state-of-the-art algorithm for the fixed-budget BAI problem [1], in which the goal is to identify the best arm with the smallest probability of error based on a given budget of draws. The proposed SHOT (Sequential Halving applied tO Trees) algorithm [6] is compared empirically to the UCT approach of [17], showing improvements in some cases. A hybrid approach mixing SHOT and UCT is also studied [19], still without sample complexity guarantees.

In the fixed-confidence setting, [22] develop the first sample complexity guarantees in the model we consider. The proposed algorithm, FindTopWinner is based on *uniform sampling and eliminations*, an approach that may be related to the Successive Eliminations algorithm [7] for fixed-confidence BAI in bandit models. FindTopWinner proceeds in rounds, in which the leaves that have not been eliminated are sampled repeatedly until the precision of their estimates doubled. Then the tree is pruned of every node whose estimated value differs significantly from the estimated value of its parent, which leads to the possible elimination of several leaves. For depth-two trees, [10] propose an elimination procedure that is not round-based. In this simpler setting, an algorithm that exploits *confidence intervals* is also developed, inspired by the LUCB algorithm for fixed-confidence BAI [13]. Some variants of the proposed M-LUCB algorithm appear to perform better in simulations than elimination based algorithms. We now investigate this trend further in deeper trees, both in theory and in practice.

**Our Contribution.** In this paper, we propose a generic architecture, called BAI-MCTS, that builds on a Best Arm Identification (BAI) algorithm and on confidence intervals on the node values in order to solve the best action identification problem in a tree of arbitrary depth. In particular, we study two specific instances, UGapE-MCTS and LUCB-MCTS, that rely on *confidence-based* BAI algorithms [8, 13]. We prove that these are $(\epsilon, \delta)$-correct and give a high-probability upper bound on their sample complexity. Both our theoretical and empirical results improve over the elimination-based state-of-the-art algorithm, FindTopWinner [22].

## 2   BAI-MCTS algorithms

We present a generic class of algorithms, called BAI-MCTS, that combines a BAI algorithm with an exploration of the tree based on confidence intervals on the node values. Before introducing the algorithm and two particular instances, we first explain how to build such confidence intervals, and also introduce the central notion of *representative child* and *representative leaf*.

### 2.1   Confidence intervals and representative nodes

For each leaf $\ell \in \mathcal{L}$, using the past observations from this leaf we may build a confidence interval
$$\mathcal{I}_\ell(t) = [\mathrm{L}_\ell(t), \mathrm{U}_\ell(t)],$$
where $\mathrm{U}_\ell(t)$ (resp. $\mathrm{L}_\ell(t)$) is an Upper Confidence Bound (resp. a Lower Confidence Bound) on the value $V(\ell) = \mu_\ell$. The specific confidence interval we shall use will be discussed later.

These confidence intervals are then propagated upwards in the tree using the following construction. For each internal node $s$, we recursively define $\mathcal{I}_s(t) = [\mathrm{L}_s(t), \mathrm{U}_s(t)]$ with

$$\mathrm{L}_s(t) = \begin{cases} \max_{c \in \mathcal{C}(s)} \mathrm{L}_c(t) & \text{for a MAX node } s, \\ \min_{c \in \mathcal{C}(s)} \mathrm{L}_c(t) & \text{for a MIN node } s, \end{cases} \quad \mathrm{U}_s(t) = \begin{cases} \max_{c \in \mathcal{C}(s)} \mathrm{U}_c(t) & \text{for a MAX node } s, \\ \min_{c \in \mathcal{C}(s)} \mathrm{U}_c(t) & \text{for a MIN node } s. \end{cases}$$

Note that these intervals are the tightest possible on the parent under the sole assumption that the child confidence intervals are all valid. A similar construction was used in the OMS algorithm of [3] in a different context. It is easy to convince oneself (or prove by induction, see Appendix B.1) that the accuracy of the confidence intervals is preserved under this construction, as stated below.

**Proposition 1.** *Let $t \in \mathbb{N}$. One has $\bigcap_{\ell \in \mathcal{L}} (\mu_\ell \in \mathcal{I}_\ell(t)) \Rightarrow \bigcap_{s \in \mathcal{T}} (V_s \in \mathcal{I}_s(t))$.*

We now define the *representative child* $c_s(t)$ of an internal node $s$ as
$$c_s(t) = \begin{cases} \operatorname{argmax}_{c \in \mathcal{C}(s)} \mathrm{U}_c(t) & \text{if } s \text{ is a MAX node}, \\ \operatorname{argmin}_{c \in \mathcal{C}(s)} \mathrm{L}_c(t) & \text{if } s \text{ is a MIN node}, \end{cases}$$
and the *representative leaf* $\ell_s(t)$ of a node $s \in \mathcal{T}$, which is the leaf obtained when going down the tree by always selecting the representative child:
$$\ell_s(t) = s \text{ if } s \in \mathcal{L}, \quad \ell_s(t) = \ell_{c_s(t)}(t) \text{ otherwise.}$$

The confidence intervals in the tree represent the statistically plausible values in each node, hence the representative child can be interpreted as an "optimistic move" in a MAX node and a "pessimistic move" in a MIN node (assuming we play against the best possible adversary). This is reminiscent of the behavior of the UCT algorithm [17]. The construction of the confidence intervals and associated representative children are illustrated in Figure 1.

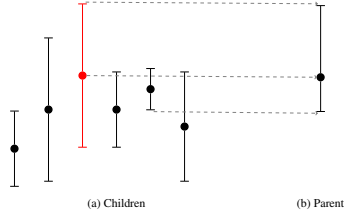

(a) Children       (b) Parent

Figure 1: Construction of confidence interval and representative child (in red) for a MAX node.

**Input**: a BAI algorithm
**Initialization**: $t = 0$.
**while** *not* $\text{BAIStop}\left(\{s \in \mathcal{C}(s_0)\}\right)$ **do**
    $R_{t+1} = \text{BAIStep}\left(\{s \in \mathcal{C}(s_0)\}\right)$
    Sample the representative leaf
      $L_{t+1} = \ell_{R_{t+1}}(t)$
    Update the information about the arms.
    $t = t + 1$.
**end**
**Output**: $\text{BAIReco}\left(\{s \in \mathcal{C}(s_0)\}\right)$

Figure 2: The BAI-MCTS architecture

## 2.2 The BAI-MCTS architecture

In this section we present the generic BAI-MCTS algorithm, whose sampling rule combines two ingredients: a best arm identification step which selects an action at the root, followed by a *confidence based exploration step*, that goes down the tree starting from this depth-one node in order to select the *representative leaf* for evaluation.

The structure of a BAI-MCTS algorithm is presented in Figure 2. The algorithm depends on a Best Arm Identification (BAI) algorithm, and uses the three components of this algorithm:

- the sampling rule $\text{BAIStep}(\mathcal{S})$ selects an arm in the set $\mathcal{S}$
- the stopping rule $\text{BAIStop}(\mathcal{S})$ returns `True` if the algorithm decides to stop
- the recommendation rule $\text{BAIReco}(\mathcal{S})$ selects an arm as a candidate for the best arm

In BAI-MCTS, the arms are the depth-one nodes, hence the information needed by the BAI algorithm to make a decision (e.g. `BAIStep` for choosing an arm, or `BAIStop` for stopping) is information about depth-one nodes, that has to be updated at the end of each round (last line in the `while` loop). Different BAI algorithms may require different information, and we now present two instances that rely on confidence intervals (and empirical estimates) for the value of the depth-one nodes.

## 2.3 UGapE-MCTS and LUCB-MCTS

Several Best Arm Identification algorithms may be used within BAI-MCTS, and we now present two variants, that are respectively based on the UGapE [8] and the LUCB [13] algorithms. These two algorithms are very similar in that they exploit confidence intervals and use the same stopping rule, however the LUCB algorithm additionally uses the empirical means of the arms, which within BAI-MCTS requires defining an estimate $\hat{V}_s(t)$ of the value of the depth-one nodes.

The generic structure of the two algorithms is similar. At round $t + 1$ two promising depth-one nodes are computed, that we denote by $\underline{b}_t$ and $\underline{c}_t$. Among these two candidates, the node whose confidence interval is the largest (that is, the most uncertain node) is selected:

$$R_{t+1} = \underset{i \in \{\underline{b}_t, \underline{c}_t\}}{\text{argmax}} \left[ U_i(t) - L_i(t) \right].$$

Then, following the BAI-MCTS architecture, the representative leaf of $R_{t+1}$ (computed by going down the tree) is sampled: $L_{t+1} = \ell_{R_{t+1}}(t)$. The algorithm stops whenever the confidence intervals of the two promising arms overlap by less than $\epsilon$:

$$\tau = \inf \left\{ t \in \mathbb{N} : U_{\underline{c}_t}(t) - L_{\underline{b}_t}(t) < \epsilon \right\},$$

and it recommends $\hat{s}_\tau = \underline{b}_\tau$.

In both algorithms that we detail below $\underline{b}_t$ represents a guess for the best depth-one node, while $\underline{c}_t$ is an "optimistic" challenger, that has the maximal possible value among the other depth-one nodes. Both nodes need to be explored enough in order to discover the best depth-one action quickly.

**UGapE-MCTS.** In UGapE-MCTS, introducing for each depth-one node the index

$$B_s(t) = \max_{s' \in \mathcal{C}(s_0)\setminus\{s\}} U_{s'}(t) - L_s(t),$$

the promising depth-one nodes are defined as

$$\underline{b}_t = \operatorname*{argmin}_{a \in \mathcal{C}(s_0)} B_a(t) \quad \text{and} \quad \underline{c}_t = \operatorname*{argmax}_{b \in \mathcal{C}(s_0)\setminus\{\underline{b}_t\}} U_b(t).$$

**LUCB-MCTS.** In LUCB-MCTS, the promising depth-one nodes are defined as

$$\underline{b}_t = \operatorname*{argmax}_{a \in \mathcal{C}(s_0)} \hat{V}_a(t) \quad \text{and} \quad \underline{c}_t = \operatorname*{argmax}_{b \in \mathcal{C}(s_0)\setminus\{\underline{b}_t\}} U_b(t),$$

where $\hat{V}_s(t) = \hat{\mu}_{\ell_s(t)}(t)$ is the empirical mean of the reprentative leaf of node $s$. Note that several alternative definitions of $\hat{V}_s(t)$ may be proposed (such as the middle of the confidence interval $\mathcal{I}_s(t)$, or $\max_{a \in \mathcal{C}(s)} \hat{V}_a(t)$), but our choice is crucial for the analysis of LUCB-MCTS, given in Appendix C.

## 3 Analysis of UGapE-MCTS

In this section we first prove that UGapE-MCTS and LUCB-MCTS are both $(\epsilon, \delta)$-correct. Then we give in Theorem 3 a high-probability upper bound on the number of samples used by UGapE-MCTS. A similar upper bound is obtained for LUCB-MCTS in Theorem 9, stated in Appendix C.

### 3.1 Choosing the Confidence Intervals

From now on, we assume that the confidence intervals on the leaves are of the form

$$L_\ell(t) = \hat{\mu}_\ell(t) - \sqrt{\frac{\beta(N_\ell(t), \delta)}{2N_\ell(t)}} \quad \text{and} \quad U_\ell(t) = \hat{\mu}_\ell(t) + \sqrt{\frac{\beta(N_\ell(t), \delta)}{2N_\ell(t)}}. \tag{1}$$

$\beta(s, \delta)$ is some *exploration function*, that can be tuned to have a $\delta$-PAC algorithm, as expressed in the following lemma, whose proof can be found in Appendix B.2

**Lemma 2.** *If $\delta \le \max(0.1|\mathcal{L}|, 1)$, for the choice*

$$\beta(s, \delta) = \ln(|\mathcal{L}|/\delta) + 3\ln\ln(|\mathcal{L}|/\delta) + (3/2)\ln(\ln s + 1) \tag{2}$$

*both UGapE-MCTS and LUCB-MCTS satisfy $\mathbb{P}(V(s^*) - V(\hat{s}_\tau) \le \epsilon) \ge 1 - \delta$.*

An interesting practical feature of these confidence intervals is that they only depend on the local number of draws $N_\ell(t)$, whereas most of the BAI algorithms use exploration functions that depend on the number of rounds $t$. Hence the only confidence intervals that need to be updated at round $t$ are those of the ancestors of the selected leaf, which can be done recursively.

Moreover, $\beta(s, \delta)$ scales with $\ln(\ln(s))$, and not $\ln(s)$, leveraging some tools recently introduced to obtain tighter confidence intervals [12, 15]. The union bound over $\mathcal{L}$ (that may be an artifact of our current analysis) however makes the exploration function of Lemma 2 still a bit over-conservative and in practice, we recommend the use of $\beta(s, \delta) = \ln(\ln(es)/\delta)$.

Finally, similar correctness results (with slightly larger exploration functions) may be obtained for confidence intervals based on the Kullback-Leibler divergence (see [5]), which are known to lead to better performance in standard best arm identification problems [16] and also depth-two tree search problems [10]. However, the sample complexity analysis is much more intricate, hence we stick to the above Hoeffding-based confidence intervals for the next section.

### 3.2 Complexity term and sample complexity guarantees

We first introduce some notation. Recall that $s^*$ is the optimal action at the root, identified with the depth-one node satisfying $V(s^*) = V(s_0)$, and define the second-best depth-one node as $s_2^* =$

$\text{argmax}_{s \in \mathcal{C}(s_0) \setminus \{s^*\}} V_s$. Recall $\mathcal{P}(s)$ denotes the parent of a node $s$ different from the root. Introducing furthermore the set $\text{Anc}(s)$ of all the ancestors of a node $s$, we define the complexity term by

$$H_\epsilon^*(\boldsymbol{\mu}) := \sum_{\ell \in \mathcal{L}} \frac{1}{\Delta_\ell^2 \vee \Delta_*^2 \vee \epsilon^2}, \quad \text{where} \quad \begin{array}{rcl} \Delta_* & := & V(s^*) - V(s_2^*) \\ \Delta_\ell & := & \max_{s \in \text{Anc}(\ell) \setminus \{s_0\}} |V_s - V(\mathcal{P}(s))| \end{array} \tag{3}$$

The intuition behind these squared terms in the denominator is the following. We will sample a leaf $\ell$ until we either prune it (by determining that it or one of its ancestors is a bad move), prune everyone else (this happens for leaves below the optimal arm) or reach the required precision $\epsilon$.

**Theorem 3.** *Let $\delta \le \min(1, 0.1|\mathcal{L}|)$. UGapE-MCTS using the exploration function* (2) *is such that, with probability larger than $1 - \delta$, $(V(s^*) - V(\hat{s}_\tau) < \epsilon)$ and, letting $\overline{\Delta}_{\ell,\epsilon} = \Delta_\ell \vee \Delta_* \vee \epsilon$,*

$$\tau \le 8H_\epsilon^*(\boldsymbol{\mu}) \ln \frac{|\mathcal{L}|}{\delta} + \sum_\ell \frac{16}{\overline{\Delta}_{\ell,\epsilon}^2} \ln \ln \frac{1}{\overline{\Delta}_{\ell,\epsilon}^2}$$

$$+ \ 8H_\epsilon^*(\boldsymbol{\mu}) \left[ 3 \ln \ln \frac{|\mathcal{L}|}{\delta} + 2 \ln \ln \left( 8e \ln \frac{|\mathcal{L}|}{\delta} + 24e \ln \ln \frac{|\mathcal{L}|}{\delta} \right) \right] + 1.$$

**Remark 4.** *If $\beta(N_a(t), \delta)$ is changed to $\beta(t, \delta)$, one can still prove $(\epsilon, \delta)$ correctness and furthermore upper bound the expectation of $\tau$. However the algorithm becomes less efficient to implement, since after each leaf observation, ALL the confidence intervals have to be updated. In practice, this change lowers the probability of error but does not effect significantly the number of play-outs used.*

### 3.3 Comparison with previous work

To the best of our knowledge[1], the $\text{FindTopWinner}$ algorithm [22] is the only algorithm from the literature designed to solve the best action identification problem in any-depth trees. The number of play-outs of this algorithm is upper bounded with high probability by

$$\sum_{\ell: \Delta_\ell > 2\epsilon} \left( \frac{32}{\Delta_\ell^2} \ln \frac{16|\mathcal{L}|}{\Delta_\ell \delta} + 1 \right) + \sum_{\ell: \Delta_\ell \le 2\epsilon} \left( \frac{8}{\epsilon^2} \ln \frac{8|\mathcal{L}|}{\epsilon \delta} + 1 \right)$$

One can first note the improvement in the constant in front of the leading term in $\ln(1/\delta)$, as well as the presence of the $\ln \ln(1/\overline{\Delta}_{\ell,\epsilon^2})$ second order, that is unavoidable in a regime in which the gaps are small [12]. The most interesting improvement is in the control of the number of draws of $2\epsilon$-optimal leaves (such that $\Delta_\ell \le 2\epsilon$). In UGapE-MCTS, the number of draws of such leaves is at most of order $(\epsilon \vee \Delta_*^2)^{-1} \ln(1/\delta)$, which may be significantly smaller than $\epsilon^{-1} \ln(1/\delta)$ if there is a gap in the best and second best value. Moreover, unlike $\text{FindTopWinner}$ and M-LUCB [10] in the depth two case, UGapE-MCTS can also be used when $\epsilon = 0$, with provable guarantees.

Regarding the algorithms themselves, one can note that M-LUCB, an extension of LUCB suited for depth-two tree, does *not* belong to the class of BAI-MCTS algorithms. Indeed, it has a "reversed" structure, first computing the representative leaf for each depth-one node: $\forall s \in \mathcal{C}(s_0), R_{s,t} = \ell_s(t)$ and then performing a BAI step over the representative leaves: $\tilde{L}_{t+1} = \text{BAIStep}(R_{s,t}, s \in \mathcal{C}(s_0))$. This alternative architecture can also be generalized to deeper trees, and was found to have empirical performance similar to BAI-MCTS. M-LUCB, which will be used as a benchmark in Section 4, also distinguish itself from LUCB-MCTS by the fact that it uses an exploration rate that depends on the global time $\beta(t, \delta)$ and that $\underline{b}_t$ is the empirical maximin arm (which can be different from the arm maximizing $\hat{V}_s$). This alternative choice is not yet supported by theoretical guarantees in deeper trees.

Finally, the exploration step of BAI-MCTS algorithm bears some similarity with the UCT algorithm [17], as it goes down the tree choosing alternatively the move that yields the highest UCB or the lowest LCB. However, the behavior of BAI-MCTS is very different at the root, where the first move is selected using a BAI algorithm. Another key difference is that BAI-MCTS relies on *exact* confidence

intervals: each interval $\mathcal{I}_s(t)$ is shown to contain with high probability the corresponding value $V_s$, whereas UCT uses more heuristic confidence intervals, based on the number of visits of the parent node, and aggregating all the samples from descendant nodes. Using UCT in our setting is not obvious as it would require to define a suitable stopping rule, hence we don't include a comparison with this algorithm in Section 4. A hybrid comparison between UCT and FindTopWinner is proposed in [22], providing UCT with the random number of samples used by the the fixed-confidence algorithm. It is shown that FindTopWinner has the advantage for hard trees that require many samples. Our experiments show that our algorithms in turn always dominate FindTopWinner.

### 3.4 Proof of Theorem 3.

Letting $\mathcal{E}_t = \bigcap_{\ell \in \mathcal{L}} (\mu_\ell \in \mathcal{I}_\ell(t))$ and $\mathcal{E} = \bigcap_{t \in \mathbb{N}} \mathcal{E}_t$, we upper bound $\tau$ assuming the event $\mathcal{E}$ holds, using the following key result, which is proved in Appendix D.

**Lemma 5.** *Let $t \in \mathbb{N}$. $\mathcal{E}_t \cap (\tau > t) \cap (L_{t+1} = \ell) \implies N_\ell(t) \le \frac{8\beta(N_\ell(t), \delta)}{\Delta_\ell^2 \vee \Delta_*^2 \vee \epsilon^2}$.*

An intuition behind this result is the following. First, using that the selected leaf $\ell$ is a representative leaf, it can be seen that the confidence intervals from $s_D = \ell$ to $s_0$ are nested (Lemma 11). Hence if $\mathcal{E}_t$ holds, $V(s_k) \in \mathcal{I}_\ell(t)$ for all $k = 1, \ldots, D$, which permits to lower bound the width of this interval (and thus upper bound $N_\ell(t)$) as a function of the $V(s_k)$ (Lemma 12). Then Lemma 13 exploits the mechanism of UGapE to further relate this width to $\Delta_*$ and $\epsilon$.

Another useful tool is the following lemma, that will allow to leverage the particular form of the exploration function $\beta$ to obtain an explicit upper bound on $N_\ell(\tau)$.

**Lemma 6.** *Let $\beta(s) = C + \frac{3}{2}\ln(1 + \ln(s))$ and define $S = \sup\{s \ge 1 : a\beta(s) \ge s\}$. Then*

$$S \le aC + 2a \ln(1 + \ln(aC)).$$

This result is a consequence of Theorem 16 stated in Appendix F, that uses the fact that for $C \ge -\ln(0.1)$ and $a \ge 8$, it holds that

$$\frac{3}{2} \frac{C(1 + \ln(aC))}{C(1 + \ln(aC)) - \frac{3}{2}} \le 1.7995564 \le 2.$$

On the event $\mathcal{E}$, letting $\tau_\ell$ be the last instant before $\tau$ at which the leaf $\ell$ has been played before stopping, one has $N_\ell(\tau - 1) = N_\ell(\tau_\ell)$ that satisfies by Lemma 5

$$N_\ell(\tau_\ell) \le \frac{8\beta(N_\ell(\tau_\ell), \delta)}{\Delta_\ell^2 \vee \Delta_*^2 \vee \epsilon^2}.$$

Applying Lemma 6 with $a = a_\ell = \frac{8}{\Delta_\ell^2 \vee \Delta_*^2 \vee \epsilon^2}$ and $C = \ln \frac{|\mathcal{L}|}{\delta} + 3\ln\ln \frac{|\mathcal{L}|}{\delta}$ leads to

$$N_\ell(\tau - 1) \le a_\ell (C + 2\ln(1 + \ln(a_\ell C))).$$

Letting $\overline{\Delta}_{\ell,\epsilon} = \Delta_\ell \vee \Delta_* \vee \epsilon$ and summing over arms, we find

$$\tau = 1 + \sum_\ell N_\ell(\tau - 1)$$

$$\le 1 + \sum_\ell \frac{8}{\overline{\Delta}_{\ell,\epsilon}^2} \left( \ln \frac{|\mathcal{L}|}{\delta} + 3\ln\ln \frac{|\mathcal{L}|}{\delta} + 2\ln\ln \left( 8e \frac{\ln \frac{|\mathcal{L}|}{\delta} + 3\ln\ln \frac{|\mathcal{L}|}{\delta}}{\overline{\Delta}_{\ell,\epsilon}^2} \right) \right)$$

$$= 1 + \sum_\ell \frac{8}{\overline{\Delta}_{\ell,\epsilon}^2} \left( \ln \frac{|\mathcal{L}|}{\delta} + 2\ln\ln \frac{1}{\overline{\Delta}_{\ell,\epsilon}^2} \right) + 8H_\epsilon^*(\boldsymbol{\mu}) \left[ 3\ln\ln \frac{|\mathcal{L}|}{\delta} + 2\ln\ln \left( 8e\ln \frac{|\mathcal{L}|}{\delta} + 24e\ln\ln \frac{|\mathcal{L}|}{\delta} \right) \right].$$

To conclude the proof, we remark that from the proof of Lemma 2 (see Appendix B.2) it follows that on $\mathcal{E}$, $V(s^*) - V(\hat{s}_\tau) < \epsilon$ and that $\mathcal{E}$ holds with probability larger than $1 - \delta$.

## 4 Experimental Validation

In this section we evaluate the performance of our algorithms in three experiments. We evaluate on the depth-two benchmark tree from [10], a new depth-three tree and the random tree ensemble from [22]. We compare to the $\mathrm{FindTopWinner}$ algorithm from [22] in all experiments, and in the depth-two experiment we include the M-LUCB algorithm from [10]. Its relation to BAI-MCTS is discussed in Section 3.3. For our BAI-MCTS algorithms and for M-LUCB we use the exploration rate $\beta(s, \delta) = \ln \frac{|\mathcal{L}|}{\delta} + \ln(\ln(s) + 1)$ (a stylized version of Lemma 2 that works well in practice), and we use the KL refinement of the confidence intervals (1). To replicate the experiment from [22], we supply all algorithms with $\delta = 0.1$ and $\epsilon = 0.01$. For comparing with [10] we run all algorithms with $\epsilon = 0$ and $\delta = 0.1|\mathcal{L}|$ (undoing the conservative union bound over leaves. This excessive choice, which might even exceed one, does not cause a problem, as the algorithms depend on $\frac{\delta}{|\mathcal{L}|} = 0.1$). In none of our experiments the observed error rate exceeds $0.1$.

Figure 3 shows the benchmark tree from [10, Section 5] and the performance of four algorithms on it. We see that the special-purpose depth-two M-LUCB performs best, very closely followed by both our new arbitrary-depth LUCB-MCTS and UGapE-MCTS methods. All three use significantly fewer samples than $\mathrm{FindTopWinner}$. Figure 4 (displayed in Appendix A for the sake of readability) shows a full 3-way tree of depth 3 with leafs drawn uniformly from $[0, 1]$. Again our algorithms outperform the previous state of the art by an order of magnitude. Finally, we replicate the experiment from [22, Section 4]. To make the comparison as fair as possible, we use the proven exploration rate from (2). On 10K full 10-ary trees of depth 3 with Bernoulli leaf parameters drawn uniformly at random from $[0, 1]$ the average numbers of samples are: LUCB-MCTS 141811, UGapE-MCTS 142953 and FindTopWinner 2254560. To closely follow the original experiment, we do apply the union bound over leaves to all algorithms, which are run with $\epsilon = 0.01$ and $\delta = 0.1$. We did not observe any error from any algorithm (even though we allow 10%). Our BAI-MCTS algorithms deliver an impressive 15-fold reduction in samples.

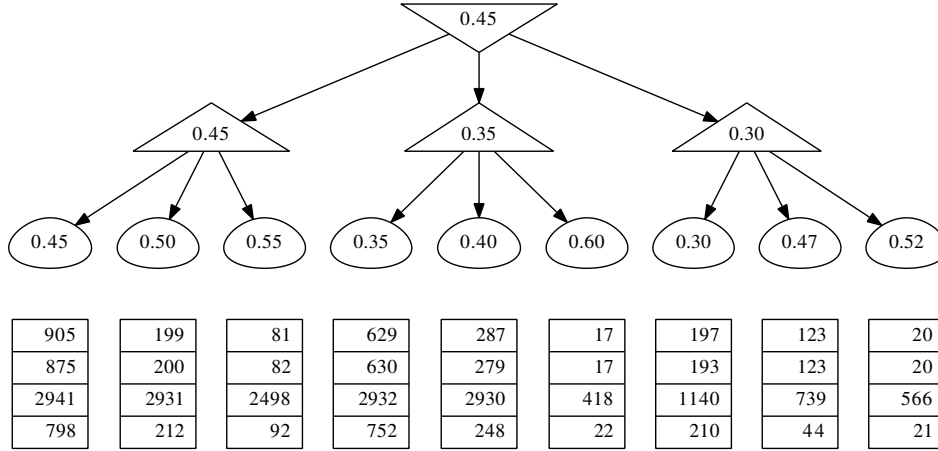

Figure 3: The $3 \times 3$ tree of depth 2 that is the benchmark in [10]. Shown below the leaves are the average numbers of pulls for 4 algorithms: LUCB-MCTS (0.89% errors, 2460 samples), UGapE-MCTS (0.94%, 2419), $\mathrm{FindTopWinner}$ (0%, 17097) and M-LUCB (0.14%, 2399). All counts are averages over 10K repetitions with $\epsilon = 0$ and $\delta = 0.1 \cdot 9$.

## 5 Lower bounds and discussion

Given a tree $\mathcal{T}$, a MCTS model is parameterized by the leaf values, $\boldsymbol{\mu} := (\mu_\ell)_{\ell \in \mathcal{L}}$, which determine the best root action: $s^* = s^*(\boldsymbol{\mu})$. For $\boldsymbol{\mu} \in [0, 1]^{|\mathcal{L}|}$, We define $\mathrm{Alt}(\boldsymbol{\mu}) = \{\boldsymbol{\lambda} \in [0, 1]^{|\mathcal{L}|} : s^*(\boldsymbol{\lambda}) \neq s^*(\boldsymbol{\mu})\}$. Using the same technique as [9] for the classic best arm identification problem, one can establish the following (non explicit) lower bound. The proof is given in Appendix E.

**Theorem 7.** *Assume $\epsilon = 0$. Any $\delta$-correct algorithm satisfies*

$$\mathbb{E}_{\boldsymbol{\mu}}[\tau] \geq T^*(\boldsymbol{\mu})d(\delta, 1-\delta), \quad where \quad T^*(\boldsymbol{\mu})^{-1} := \sup_{\boldsymbol{w} \in \Sigma_{|\mathcal{L}|}} \inf_{\boldsymbol{\lambda} \in \mathrm{Alt}(\boldsymbol{\mu})} \sum_{\ell \in \mathcal{L}} w_\ell d(\mu_\ell, \lambda_\ell) \tag{4}$$

*with $\Sigma_k = \{\boldsymbol{w} \in [0,1]^i : \sum_{i=1}^k w_i = 1\}$ and $d(x, y) = x \ln(x/y) + (1-x)\ln((1-x)/(1-y))$ is the binary Kullback-Leibler divergence.*

This result is however not directly amenable for comparison with our upper bounds, as the optimization problem defined in Lemma 7 is not easy to solve. Note that $d(\delta, 1-\delta) \geq \ln(1/(2.4\delta))$ [15], thus our upper bounds have the right dependency in $\delta$. For depth-two trees with $K$ (resp. $M$) actions for player A (resp. B), we can moreover prove the following result, that suggests an intriguing behavior.

**Lemma 8.** *Assume $\epsilon = 0$ and consider a tree of depth two with $\boldsymbol{\mu} = (\mu_{i,j})_{1 \leq i \leq K, 1 \leq j \leq M}$ such that $\forall (i,j), \mu_{1,1} > \mu_{i,1}, \mu_{1,1} < \mu_{i,j}$. The supremum in the definition of $T^*(\boldsymbol{\mu})^{-1}$ can be restricted to*

$$\tilde{\Sigma}_{K,M} := \{\boldsymbol{w} \in \Sigma_{K \times M} : w_{i,j} = 0 \text{ if } i \geq 2 \text{ and } j \geq 2\}$$

*and*

$$T^*(\boldsymbol{\mu})^{-1} = \max_{\boldsymbol{w} \in \tilde{\Sigma}_{K,M}} \min_{\substack{i=2,\dots,K \\ a=1,\dots,M}} \left[ w_{1,a} d\left(\mu_{1,a}, \frac{w_{1,a}\mu_{1,a} + w_{i,1}\mu_{i,1}}{w_{1,a} + w_{i,1}}\right) + w_{i,1} d\left(\mu_{i,1}, \frac{w_{1,a}\mu_{1,a} + w_{i,1}\mu_{i,1}}{w_{1,a} + w_{i,1}}\right) \right].$$

It can be extracted from the proof of Theorem 7 (see Appendix E) that the vector $\boldsymbol{w}^*(\boldsymbol{\mu})$ that attains the supremum in (4) represents the average proportions of selections of leaves by any algorithm matching the lower bound. Hence, the sparsity pattern of Lemma 8 suggests that matching algorithms should draw many of the leaves *much less than $O(\ln(1/\delta))$ times*. This hints at the exciting prospect of optimal stochastic pruning, at least in the asymptotic regime $\delta \to 0$.

As an example, we numerically solve the lower bound optimization problem (which is a concave maximization problem) for $\boldsymbol{\mu}$ corresponding to the benchmark tree displayed in Figure 3 to obtain

$$T^*(\boldsymbol{\mu}) = 259.9 \quad \text{and} \quad \boldsymbol{w}^* = (0.3633, 0.1057, 0.0532), (0.3738, \boldsymbol{0}, \boldsymbol{0}), (0.1040, \boldsymbol{0}, \boldsymbol{0}).$$

With $\delta = 0.1$ we find $\mathrm{kl}(\delta, 1-\delta) = 1.76$ and the lower bound is $\mathbb{E}_{\boldsymbol{\mu}}[\tau] \geq 456.9$. We see that there is a potential improvement of at least a factor $4$.

**Future directions** An (asymptotically) optimal algorithm for BAI called Track-and-Stop was developed by [9]. It maintains the empirical proportions of draws close to $\boldsymbol{w}^*(\hat{\boldsymbol{\mu}})$, adding forced exploration to ensure $\hat{\boldsymbol{\mu}} \to \boldsymbol{\mu}$. We believe that developing this line of ideas for MCTS would result in a major advance in the quality of tree search algorithms. The main challenge is developing efficient solvers for the general optimization problem (4). For now, even the sparsity pattern revealed by Lemma 8 for depth two does not give rise to efficient solvers. We also do not know how this sparsity pattern evolves for deeper trees, let alone how to compute $\boldsymbol{w}^*(\boldsymbol{\mu})$.

**Acknowledgments.** Emilie Kaufmann acknowledges the support of the French Agence Nationale de la Recherche (ANR), under grant ANR-16-CE40-0002 (project BADASS). Wouter Koolen acknowledges support from the Netherlands Organization for Scientific Research (NWO) under Veni grant 639.021.439.

## Footnotes

[1]In a recent paper, [11] independently proposed the LUCBMinMax algorithm, that differs from UGapE-MCTS and LUCB-MCTS only by the way the best guess $\underline{b}_t$ is picked. The analysis is very similar to ours, but features some refined complexity measure, in which $\Delta_\ell$ (that is the maximal distance between *consecutive* ancestors of the leaf, see (3)) is replaced by the maximal distance between *any* ancestors of that leaf. Similar results could be obtained for our two algorithms following the same lines.

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
