[Supplementary Material]

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

## A  Numerical Results for a Depth Three Tree

The results for our experiments on a depth-three tree are displayed in Figure 4.

## B  Confidence Intervals

### B.1  Proof of Proposition 1

The proof proceeds by induction. Let the inductive hypothesis be $\mathcal{H}_d=$"for all the nodes $s$ at (graph) distance $d$ from a leaf, $V_s \in \mathcal{I}_s(t)$".

$\mathcal{H}_0$ clearly holds by definition of $\mathcal{E}_t$. Now let $d$ such that $\mathcal{H}_d$ holds and let $s$ be at distance $d+1$ of a leaf. Then all $s' \in \mathcal{C}(s)$ are at distance at most $d$ from a leaf and using the inductive hypothesis,

$$\forall c \in \mathcal{C}(s), \quad \mathrm{L}_c(t) \le V_c \le \mathrm{U}_c(t).$$

Assume that $s$ is a MAX node. Using that $\mathrm{U}_s(t) = \max_{c \in \mathcal{C}(s)} \mathrm{U}_c(t)$, one has $c \in \mathcal{C}(s)$, $V_c \le \mathrm{U}_c(t) \le \mathrm{U}_s(t)$. By definition, $V_s = \max_{c \in \mathcal{C}(s)} V_c$, thus it follows that $V_s \le \mathrm{U}_s(t)$. Still by definition of $V_s$, it holds that $\forall c \in \mathcal{C}(s), \mathrm{L}_c(t) \le V_c \le V_s$ and finally, as $\mathrm{L}_s(t) = \max_{c \in \mathcal{C}(s)} \mathrm{L}_c(t)$, $\mathrm{L}_s(t) \le V_s \le \mathrm{U}_s(t)$. A similar reasoning yields the same conclusion if $s$ is a MIN node, thus $\mathcal{H}_{d+1}$ holds.

As the tree $\mathcal{T}$ is finite, we conclude by induction that $\forall s \in \mathcal{T}, V_s \in \mathcal{I}_s(t)$.

### B.2  Proof of Lemma 2

Let

$$\mathcal{E}_t = \bigcap_{\ell \in \mathcal{L}} (\mu_\ell \in \mathcal{I}_\ell(t)) \quad \text{and} \quad \mathcal{E} = \bigcap_{t \in \mathbb{N}} \mathcal{E}_t.$$

Using Proposition 1, on $\mathcal{E}_t$, for all $s \in \mathcal{T}$, $V_s \in \mathcal{I}_s(t)$. If the algorithm stops at some time $t$, as $\mathrm{L}_{\underline{b}_t}(t) > \mathrm{U}_{\underline{c}_t}(t) - \epsilon$, the outputted action, $\hat{s}_\tau = \underline{b}_t$, satisfies $\mathrm{L}_{\hat{s}_\tau}(t) > \mathrm{U}_{s'}(t) - \epsilon$, for all $s' \ne \hat{s}_\tau$. As $\mathcal{E}$ holds, one obtains

$$V(\hat{s}_\tau) \ge \max_{s' \ne \hat{s}_\tau} V(s') - \epsilon,$$

and $\hat{s}_\tau$ is an $\epsilon$-maximin action. Hence, the algorithm is correct on $\mathcal{E}$. The error probability is thus upper bounded by

$$
\begin{aligned}
\mathbb{P}(\mathcal{E}^c) &\le \mathbb{P}\left(\exists \ell \in \mathcal{L}, \exists t \in \mathbb{N} : |\hat{\mu}_\ell(t) - \mu_\ell| > \sqrt{\beta(N_\ell(t), \delta)/(2N_\ell(t))}\right) \\
&\le \sum_{\ell \in \mathcal{L}} \mathbb{P}\left(\exists s \in \mathbb{N} : 2s(\hat{\mu}_{\ell,s} - \mu_\ell)^2 > \beta(s, \delta)\right) \\
&\le 2|\mathcal{L}|\mathbb{P}\left(\exists s \in \mathbb{N} : S_s > \sqrt{2\sigma^2 s \beta(s, \delta)}\right),
\end{aligned}
$$

where $S_s = X_1 + \cdots + X_s$ is a martingale with $\sigma^2$-subgaussian increments, with $\sigma^2 = 1/4$. It was shown in [15] that for $\delta \le 0.1$, if

$$\beta(t, \delta) = \ln\left(\frac{1}{\delta}\right) + 3 \ln \ln\left(\frac{1}{\delta}\right) + (3/2) \ln(\ln(s) + 1),$$

one has $\mathbb{P}\left(\exists s \in \mathbb{N} : S_s > \sqrt{2\sigma^2 s \beta(s, \delta)}\right) \le \delta$, which concludes the proof.

## C  Sample complexity analysis of LUCB-MCTS

We provide an analysis of a slight variant of LUCB-MCTS that may stop at even rounds only and for $t \in 2\mathbb{N}$ draws the representative leaf of the two promising depth-one nodes:

$$L_{t+1} = \ell_{\underline{b}_t}(t) \quad \text{and} \quad L_{t+2} = \ell_{\underline{c}_t}(t). \tag{5}$$

The stopping rule is then $\tau = \inf\left\{t \in 2\mathbb{N}^* : U_{\underline{c}_t}(t) - L_{\underline{b}_t}(t) < \epsilon\right\}$.

Figure 4: Our benchmark 3-way tree of depth 3. Shown below the leaves are the numbers of pulls of 3 algorithms: LUCB-MCTS (0.72% errors, 1551 samples), UGapE-MCTS (0.75%, 1584), and FindTopWinner (0%, 20730). Numbers are averages over 10K repetitions with $\epsilon = 0$ and $\delta = 0.1 \cdot 27$.

For this algorithm, our sample complexity guarantee features a slightly different complexity term. For a leaf $\ell = s_0 s_1 \ldots s_D$, we first introduce

$$\tilde{\Delta}_\ell = \max_{s \in \mathrm{Anc}(\ell) \setminus \{s_0, s_1\}} |V(s) - V(\mathbb{P}(s))|,$$

a quantity that differs from $\Delta_\ell$ only by the fact that the maximum doesn't take into account the gap between the root and the depth-one ancestor of $\ell$. Then $\tilde{H}_\epsilon^*(\boldsymbol{\mu})$ is defined similarly as $H_\epsilon^*(\boldsymbol{\mu})$ by

$$\tilde{H}_\epsilon^*(\boldsymbol{\mu}) = \sum_{\ell \in \mathcal{L}} \frac{1}{\tilde{\Delta}_\ell^2 \vee \Delta_*^2 \vee \epsilon^2}.$$

**Theorem 9.** *Let $\delta \leq \min(1, 0.1|\mathcal{L}|)$. LUCB-MCTS using the exploration function* (2) *and selecting the two promising leaves at each round is such that, with probability larger than $1 - \delta$, $(V(s^*) - V(\hat{s}_\tau) < \epsilon)$ and*

$$\tau \leq 16\tilde{H}_\epsilon^*(\boldsymbol{\mu}) \left[ \ln \frac{|\mathcal{L}|}{\delta} + 3 \ln \ln \frac{|\mathcal{L}|}{\delta} + 2 \ln \ln \left( 16e\tilde{H}_\epsilon^*(\boldsymbol{\mu}) \left( \ln \frac{|\mathcal{L}|}{\delta} + 3 \ln \ln \frac{|\mathcal{L}|}{\delta} \right) \right) \right].$$

**Proof.** The analysis follows the same lines as that of UGapE-MCTS, yet it relies on a slightly different key result, proved in the next section. Letting $\mathcal{E}_t = \cap_{\ell \in \mathcal{L}} (\mu_\ell \in \mathcal{I}_\ell(t))$ as in the proof of Theorem 3 and defining $\mathcal{E} = \cap_{t \in 2\mathbb{N}^*} \mathcal{E}_t$, one can state the following.

**Lemma 10.** *Let $t \in 2\mathbb{N}$.*

$$\mathcal{E}_t \cap (\tau > t) \quad \Rightarrow \quad \exists \ell \in \{L_{t+1}, L_{t+2}\}: \; N_\ell(t) \leq \frac{8\beta(t, \delta)}{\tilde{\Delta}_\ell^2 \vee \Delta_*^2 \vee \epsilon^2}.$$

Let $T$ be a deterministic time. We upper bound $\tau$ assuming the event $\mathcal{E}$ holds. Using Lemma 10 and the fact that for every even $t$, $(\tau_\delta > t) = (\tau_\delta > t + 1)$ by definition of the algorithm, one has

$$
\begin{aligned}
\min(\tau, T) &= \sum_{t=0}^{T} \mathbb{1}_{(\tau > t)} = 2 \sum_{\substack{t \in 2\mathbb{N} \\ t \leq T}} \mathbb{1}_{(\tau_\delta > t)} = 2 \sum_{\substack{t \in 2\mathbb{N} \\ t \leq T}} \mathbb{1}_{\left( \exists \ell \in \{L_{t+1}, L_{t+2}\}: N_\ell(t) \leq 2\beta(t, \delta)/(\tilde{\Delta}_\ell^2 \vee \Delta_*^2 \epsilon^2) \right)} \\
&\leq 2 \sum_{\substack{t \in 2\mathbb{N} \\ t \leq T}} \sum_{\ell \in \mathcal{L}} \mathbb{1}_{(L_{t+1} = \ell) \cup (L_{t+2} = \ell)} \mathbb{1}_{\left( N_\ell(t) \leq 8\beta(T, \delta)/(\tilde{\Delta}_\ell^2 \vee \Delta_*^2 \epsilon^2) \right)} \\
&\leq 16 \sum_{\ell \in \mathcal{L}} \frac{1}{\tilde{\Delta}_\ell^2 \vee \Delta_*^2 \vee \epsilon^2} \beta(T, \delta) = 16\tilde{H}_\epsilon^*(\boldsymbol{\mu}) \beta(T, \delta).
\end{aligned}
$$

For any $T$ such that $16\tilde{H}_\epsilon^*(\boldsymbol{\mu}) \beta(T, \delta) < T$, one has $\min(\tau, T) < T$, which implies $\tau < T$. Therefore

$$\tau \leq \sup\{t \in \mathbb{N} : 16\tilde{H}_\epsilon^*(\boldsymbol{\mu}) \beta(t, \delta) \geq t\}.$$

Just like in the analysis of UGapE-MCTS, the conclusion now follows from Lemma 6, applied with $a = 16\tilde{H}_\epsilon^*(\boldsymbol{\mu})$ and $C = \ln(|\mathcal{L}|/\delta) + 3 \ln \ln(|\mathcal{L}|/\delta)$, which yields

$$\tau \leq 16\tilde{H}_\epsilon^*(\boldsymbol{\mu}) \left[ \ln \frac{|\mathcal{L}|}{\delta} + 3 \ln \ln \frac{|\mathcal{L}|}{\delta} + 2 \ln \ln \left( 16e\tilde{H}_\epsilon^*(\boldsymbol{\mu}) \left( \ln \frac{|\mathcal{L}|}{\delta} + 3 \ln \ln \frac{|\mathcal{L}|}{\delta} \right) \right) \right].$$

Using that $\mathcal{P}(\mathcal{E}) \geq 1 - \delta$ and that the algorithm is correct on $\mathcal{E}$ yields the conclusion.

# D    Proof of Lemma 5 and Lemma 10

We first state Lemma 12, that holds for both UGapE and LUCB-MCTS and is a consequence of the definition of the exploration procedure. This result builds on the following lemma, that expresses the fact that along a path from the root to a representative leaf, the confidence intervals are nested.

**Lemma 11.** *Let $t \in \mathbb{N}$ and $s_0, s_1, \ldots, s_D$ be a path from the root down to a leaf $\ell = s_D$.*

$$(\ell_{s_1}(t) = s_D) \Rightarrow (\forall k = 2, \ldots, D, \quad \mathcal{I}_{s_{k-1}}(t) \subseteq \mathcal{I}_{s_k}(t))$$

**Lemma 12.** *Let $t \in \mathbb{N}$ and $s_0, s_1, \ldots, s_D$ be a path from the root down to a leaf $\ell = s_D$. If $\mathcal{E}_t$ holds and $\ell$ is selected at round $t + 1$ (UGapE) or if $t$ is even and $\ell \in \{L_{t+1}, L_{t+2}\}$ (LUCB), then*

$$\sqrt{\frac{2\beta(N_\ell(t), \delta)}{N_\ell(t)}} \geq \max_{k=2\ldots D} |V(s_k) - V(s_{k-1})|.$$

**UGapE-MCTS: proof of Lemma 5.** The following lemma is specific to UGapE-MCTS. We let $s_0, s_1, \ldots, s_D$ be a path down to a leaf $\ell = s_D$.

**Lemma 13.** *Let $t \in \mathbb{N}$. If $\mathcal{E}_t$ holds and UGapE-MCTS has not stopped after $t$ observations, that is $(\tau > t)$,*

$$(L_{t+1} = \ell) \Rightarrow \left( \sqrt{\frac{8\beta(N_\ell(t), \delta)}{N_\ell(t)}} \geq \max\left(\Delta_*, V(s_0) - V(s_1), \epsilon\right) \right)$$

Putting together Lemma 12 and Lemma 13 and using that

$$\Delta_\ell = \max\left(V(s_0) - V(s_1), \max_{k=2\ldots D} |V(s_k) - V(s_{k-1})|\right)$$

one obtains

$$\mathcal{E}_t \cap (\tau > t) \cap (L_{t+1} = \ell) \Rightarrow \left( \sqrt{\frac{8\beta(N_\ell(t), \delta)}{N_\ell(t)}} \geq \max\left(\Delta_\ell, \Delta_*, \epsilon\right) \right),$$

which yields the proof of Lemma 5 by inverting the bound.

**LUCB-MCTS: proof of Lemma 10.** The following lemma is specific to the LUCB-MCTS algorithm. It can be viewed as a generalization of Lemma 2 in [13].

**Lemma 14.** *Let $t \in 2\mathbb{N}$ and let $\gamma \in \left[V(s_2^*), V(s^*)\right]$. If $\mathcal{E}_t$ holds and LUCB-MCTS has not stopped after $t$ observations, that is $(\tau > t)$, then*

$$\exists \ell \in \{L_{t+1}, L_{t+2}\} : (\gamma \in \mathcal{I}_\ell(t)) \cap \left( \sqrt{\frac{2\beta(t, \delta)}{N_\ell(t)}} \geq \epsilon \right).$$

Choosing $\gamma = \frac{V(s^*) + V(s_2^*)}{2}$ and letting $s_\ell$ be the depth-one ancestor of $\ell$, on $\mathcal{E}_t$ it holds that $V(s_\ell) \in \mathcal{I}_\ell(t)$ (by Lemma 11) and

$$(\gamma \in \mathcal{I}_\ell(t)) \Rightarrow \left( \sqrt{\frac{2\beta(t, \delta)}{N_\ell(t)}} \geq |V(s_\ell) - \gamma| \right)$$

$$\Rightarrow \left( \sqrt{\frac{2\beta(t, \delta)}{N_\ell(t)}} \geq \frac{V(s^*) - V(s_2^*)}{2} \right).$$

Recall $\Delta_* = V(s^*) - V(s_2^*)$. By Lemma 14, on $\mathcal{E}_t \cap (\tau > t)$, there exists $\ell \in \{L_{t+1}, L_{t+2}\}$ such that

$$\sqrt{\frac{8\beta(t, \delta)}{N_\ell(t)}} \geq \max(\Delta_*, \epsilon). \tag{6}$$

Moreover, noting that for a leaf $\ell = s_0, s_1, \ldots, s_D$,

$$\tilde{\Delta}_\ell = \max_{k=2,\ldots,D} |V(s_k) - V(s_{k-1})|$$

a consequence of Lemma 12 is that for $\ell \in \{L_{t+1}, L_{t+2}\}$,

$$\sqrt{\frac{2\beta(t, \delta)}{N_\ell(t)}} \geq \tilde{\Delta}_\ell. \tag{7}$$

Combining (6) and (7) yields

$$\mathcal{E}_t \cap (\tau > t) \Rightarrow \exists \ell \in \{L_{t+1}, L_{t+2}\} : \sqrt{\frac{8\beta(t, \delta)}{N_\ell(t)}} \geq \max\left(\tilde{\Delta}_\ell, \Delta_*, \epsilon\right),$$

which yields the proof of Lemma 10 by inverting the bound.

## D.1  Proof of Lemma 11.

The leaf $\ell$ is the representative of the depth 1 node $s_1$, therefore the path $s_1, \ldots, s_D$ is such that $c_{s_{k-1}}(t) = s_k$ for all $k = 2, \ldots, D$. Using the way the representative are build, we now show that

$$\forall k \in \{2, \ldots, D\}, \quad \mathcal{I}_{s_{k-1}}(t) \subseteq \mathcal{I}_{s_k}(t).$$

If $s_{k-1}$ is a MAX node, $\mathrm{U}_{s_{k-1}}(t) = \mathrm{U}_{s_k}(t)$ by definition and $\mathrm{L}_{s_{k-1}}(t) = \max_{s \in \mathcal{C}(s_{k-1})} \mathrm{L}_s(t) \geq \mathrm{L}_{s_k}(t)$. Similarly, if $s_{k-1}$ is a MIN node, $\mathrm{L}_{s_{k-1}}(t) = \mathrm{L}_{s_k}(t)$ by definition and $\mathrm{U}_{s_{k-1}}(t) = \min_{s \in \mathcal{C}(s_{k-1})} \mathrm{U}_s(t) \leq \mathrm{U}_{s_k}(t)$, so that in both cases $\mathcal{I}_{s_{k-1}}(t) \subseteq \mathcal{I}_{s_k}(t)$.

## D.2  Proof of Lemma 12.

Let $\ell \in \mathcal{L}$ be a leaf that is sampled based on the information available at round $t$. In particular, as $\ell$ is a representative leaf of the depth 1 node $s_1$, the path $s_1, \ldots, s_D$ is such that $c_{s_{k-1}}(t) = s_k$ for all $k = 2, \ldots, D$. Let $k = 2, \ldots, D$. If $s_{k-1} \in \{2, \ldots, D\}$ is a MAX node, it holds by definition of the representative children that, for all $s' \in \mathcal{C}(s_{k-1})$,

$$\mathrm{U}_{s_k}(t) \geq \mathrm{U}_{s'}(t).$$

Now, from Lemma 11 one has $\mathrm{U}_\ell(t) \geq \mathrm{U}_{s_k}(t)$ and from Proposition 1 as $\mathcal{E}_t$ holds, one has

$$\forall s \in \mathcal{T}, \quad V_s \in \mathcal{I}_s(t). \tag{8}$$

Using these two ingredients yields

$$
\begin{aligned}
\mathrm{U}_\ell(t) &\geq V(s') \\
\mathrm{L}_\ell(t) + 2\sqrt{\frac{\beta(N_\ell(t), \delta)}{2 N_\ell(t)}} &\geq V(s') \\
\mathrm{L}_{s_k}(t) + 2\sqrt{\frac{\beta(N_\ell(t), \delta)}{2 N_\ell(t)}} &\geq V(s') \\
V(s_k) + 2\sqrt{\frac{\beta(N_\ell(t), \delta)}{2 N_\ell(t)}} &\geq V(s').
\end{aligned}
$$

Thus

$$\sqrt{\frac{2\beta(N_\ell(t), \delta)}{N_\ell(t)}} \geq \max_{s' \in \mathcal{C}(s_{k-1})} V(s') - V(s_k) = V(s_{k-1}) - V(s_k) \geq 0.$$

If $s_{k-1}$ is a MIN node, a similar reasoning show that

$$\sqrt{\frac{2\beta(N_\ell(t), \delta)}{N_\ell(t)}} \geq V(s_k) - V(s_{k-1}) \geq 0.$$

Putting everything together yields

$$\sqrt{\frac{2\beta(N_\ell(t), \delta)}{N_\ell(t)}} \geq \max_{k=2, \ldots, D} |V(s_k) - V(s_{k-1})|.$$

## D.3  Proof of Lemma 13.

We first prove the following intermediate result, that generalizes Lemma 4 in [8].

**Lemma 15.** *For all $t \in \mathbb{N}^*$, the following holds*

$$
\begin{aligned}
\text{if } R_{t+1} = \underline{b}_t \quad &\text{then} \quad \mathrm{U}_{\underline{c}_t}(t) \leq \mathrm{U}_{\underline{b}_t}(t) \\
\text{if } R_{t+1} = \underline{c}_t \quad &\text{then} \quad \mathrm{L}_{\underline{c}_t}(t) \leq \mathrm{L}_{\underline{b}_t}(t)
\end{aligned}
$$

*Proof.* Assume $\underline{c}_t$ is selected (i.e. $R_{t+1} = \underline{c}_t$) and $L_{\underline{c}_t}(t) > L_{\underline{b}_t}(t)$. As the confidence interval on $V(\underline{c}_t)$ is larger than the confidence intervals on $V(\underline{b}_t)$ ($\underline{c}_t$ is selected), this also yields $U_{\underline{c}_t}(t) > U_{\underline{b}_t}(t)$. Hence

$$B_{\underline{b}_t}(t) = U_{\underline{c}_t}(t) - L_{\underline{b}_t}(t) > U_{\underline{b}_t}(t) - L_{\underline{c}_t}(t).$$

Also, by definition of $\underline{c}_t$, $U_{\underline{c}_t}(t) \geq U_b(t)$. Hence

$$B_{\underline{b}_t}(t) > \max_{b \neq \underline{c}_t} U_b(t) - L_{\underline{c}_t}(t) = B_{\underline{c}_t}(t),$$

which contradicts the definition of $\underline{b}_t$. Thus, we proved by contradiction that $L_{\underline{c}_t}(t) \geq L_{\underline{b}_t}(t)$.

A similar reasoning can be used to prove that $R_{t+1} = \underline{b}_t \implies U_{\underline{c}_t}(t) \leq U_{\underline{b}_t}(t)$.

$\square$

A simple consequence of Lemma 15 is the fact that, on $\mathcal{E}_t \cap (\tau > t)$,

$$(L_{t+1} = \ell) \implies \left( \sqrt{\frac{2\beta(N_\ell(t), \delta)}{N_\ell(t)}} > \epsilon \right). \tag{9}$$

Indeed, as the algorithm doesn't stop after $t$ rounds, it holds that $U_{\underline{c}_t}(t) - L_{\underline{b}_t}(t) > \epsilon$. If $\ell$ is the arm selected at round $t+1$, $\ell = \ell_{R_{t+1}}(t)$ and one can prove using Lemma 15 that $U_{R_{t+1}}(t) - L_{R_{t+1}}(t) > \epsilon$ (by distinguishing two cases). Finally, as $\mathcal{E}_t$ holds, by Lemma 11, $\mathcal{I}_{R_{t+1}}(t) \subseteq \mathcal{I}_\ell(t)$. Hence $U_\ell(t) - L_\ell(t) > \epsilon$, and (9) follows using the particular form of the confidence intervals.

To complete the proof, we now show that

$$(L_{t+1} = \ell) \implies \left( \sqrt{\frac{8\beta(N_\ell(t), \delta)}{N_\ell(t)}} > \max(\Delta_*, V(s_0) - V(s_1)) \right). \tag{10}$$

by distinguishing several cases.

**Case 1: $s^* \notin \mathrm{Anc}(\ell)$ and $R_{t+1} = \underline{c}_t$.** Using that the algorithm doesn't stop yields

$$L_{\underline{c}_t}(t) - U_{\underline{b}_t}(t) + 2(U_{\underline{c}_t}(t) - L_{\underline{c}_t}(t)) > \epsilon.$$

As $\mathcal{E}_t$ holds, $L_{\underline{c}_t}(t) \leq V(\underline{c}_t) = V(s_1)$ and $U_{\underline{b}_t}(t) \geq V(\underline{b}_t)$.

Therefore, if $\underline{b}_t = s^*$ it holds that

$$2(U_{\underline{c}_t}(t) - L_{\underline{c}_t}(t)) > V(s^*) - V(s_1) + \epsilon.$$

If $\underline{b}_t \neq s^*$, by definition of $\underline{c}_t$ one has

$$U_{\underline{c}_t}(t) \geq U_{s^*}(t) \geq V(s^*),$$

hence

$$\begin{aligned} L_{\underline{c}_t}(t) + (U_{\underline{c}_t}(t) - L_{\underline{c}_t}(t)) &\geq V(s^*) \\ V(s_1) + (U_{\underline{c}_t}(t) - L_{\underline{c}_t}(t)) &\geq V(s^*). \end{aligned}$$

Thus, recalling that $V(s_0) = V(s^*)$, whatever the value of $\underline{b}_t$, one obtains

$$2(U_{\underline{c}_t}(t) - L_{\underline{c}_t}(t)) \geq V(s_0) - V(s_1).$$

From Lemma 11 the width of $\mathcal{I}_{\underline{c}_t}(t)$ is upper bounded by the width of $\mathcal{I}_\ell(t)$, hence

$$2(U_\ell(t) - L_\ell(t)) \geq V(s_0) - V(s_1). \tag{11}$$

**Case 2: $s^* \notin \mathrm{Anc}(\ell)$ and $R_{t+1} = \underline{b}_t$.** As $s^* \neq \underline{b}_t$, by definition of $\underline{c}_t$ one has

$$U_{\underline{c}_t}(t) \geq U_{s^*}(t) \geq V(s^*).$$

Hence, using Lemma 15,

$$U_{\underline{b}_t}(t) - L_{\underline{b}_t}(t) \geq U_{\underline{c}_t}(t) - L_{\underline{b}_t}(t) \geq V(s^*) - V(s_1),$$

as $\mathcal{E}_t$ holds. Finally, by Lemma 11,

$$U_\ell(t) - L_\ell(t) \geq V(s_0) - V(s_1). \tag{12}$$

**Case 3:** $s^* \in \mathrm{Anc}(\ell)$ **and** $R_{t+1} = \underline{b}_t$. One has $\underline{b}_t = s^*$. Using that the algorithm doesn't stop yields

$$\mathrm{L}_{\underline{c}_t}(t) - \mathrm{U}_{\underline{b}_t}(t) + 2(\mathrm{U}_{\underline{b}_t}(t) - \mathrm{L}_{\underline{b}_t}(t)) > \epsilon.$$

As $\mathcal{E}_t$ holds, $\mathrm{U}_{\underline{b}_t}(t) \le V(s^*)$ and $\mathrm{L}_{\underline{c}_t}(t) \le V(\underline{c}_t) \le V(s_2^*)$. Therefore, if $\underline{b}_t = s^*$ it holds that

$$2(\mathrm{U}_{\underline{b}_t}(t) - \mathrm{L}_{\underline{b}_t}(t)) > V(s^*) - V(s_2^*) + \epsilon.$$

and by Lemma 11

$$2\left(\mathrm{U}_\ell(t) - \mathrm{L}_\ell(t)\right) \ge V(s^*) - V(s_2^*). \tag{13}$$

**Case 4:** $s^* \in \mathrm{Anc}(\ell)$ **and** $R_{t+1} = \underline{c}_t$. One has $\underline{c}_t = s^*$. Using Lemma 15 yields

$$\mathrm{U}_{\underline{c}_t}(t) - \mathrm{L}_{\underline{c}_t}(t) \ge \mathrm{U}_{\underline{c}_t}(t) - \mathrm{L}_{\underline{b}_t}(t) \ge V(s^*) - V(s_2^*),$$

as $\mathcal{E}_t$ holds and $V(\underline{b}_t) \le V(s_2^*)$. Finally, by Lemma 11,

$$\mathrm{U}_\ell(t) - \mathrm{L}_\ell(t) \ge V(s^*) - V(s_2^*). \tag{14}$$

Combining (11)-(14), we see that in all four cases

$$2(\mathrm{U}_\ell(t) - \mathrm{L}_\ell(t)) \ge \max(V(s^*) - V(s_2^*), V(s_0) - V(s_1)),$$

as for $s^* \notin \mathrm{Anc}(\ell)$, $V(s_0) - V(s_1) = V(s^*) - V(s_1) \ge V(s^*) - V(s_2^*)$, and for $s^* \in \mathrm{Anc}(\ell)$, $V(s_0) - V(s_1) = 0$. Using the expression of the confidence intervals and recalling that $\Delta_* = V(s^*) - V(s_2^*)$, one obtains

$$4\sqrt{\frac{\beta(N_\ell(t), \delta)}{2N_\ell(t)}} \ge \max(\Delta_*, V(s_0) - V(s_1))$$

which proves (10).

### D.4  Proof of Lemma 14.

Fix $\gamma \in [V(s_2^*), V(s^*)]$ and assume $\mathcal{E}_t \cap (\tau > t)$ holds. We assume (by contradiction) that $\gamma$ doesn't belong to $\mathcal{I}_{L_{t+1}}(t)$ nor to $\mathcal{I}_{L_{t+2}}(t)$. There are four possibilities:

- $\mathrm{L}_{L_{t+1}}(t) > \gamma$ and $\mathrm{L}_{L_{t+2}}(t) > \gamma$. As $\mathcal{E}_t$ holds and $L_{t+1}$ and $L_{t+2}$ are representative, it yields that there exists two nodes $s \in \mathcal{C}(s_0)$ such that $V_s > \gamma$, which contradicts the definition of $\gamma$.

- $\mathrm{U}_{L_{t+1}}(t) < \gamma$ and $\mathrm{U}_{L_{t+2}}(t) < \gamma$. From the definition of $\underline{c}_{t+1}$, it yields that for all $s \in \mathcal{C}(s_0)$, $\mathrm{U}_s(t) < \gamma$ and as $\mathcal{E}_t$ holds one obtains $V_s < \gamma$ for all $s \in \mathcal{C}(s_0)$, which contradicts the definition of $\gamma$.

- $\mathrm{L}_{L_{t+1}}(t) > \gamma$ and $\gamma > \mathrm{U}_{L_{t+2}}(t)$. This implies that $\mathrm{L}_{L_{t+1}}(t) > \mathrm{U}_{L_{t+2}}(t)$ and that $\mathrm{L}_{\underline{b}_t}(t) > \mathrm{U}_{\underline{c}_t}(t)$ (by Lemma 11 and the fact that $L_{t+1}$ and $L_{t+2}$ are representative leaves). This yields $(\tau \le t)$ and a contradiction.

- $\mathrm{U}_{L_{t+1}}(t) < \gamma$ and $\gamma < \mathrm{L}_{L_{t+2}}(t)$. This implies in particular that $\hat{\mu}_{L_{t+1}}(t) < \hat{\mu}_{L_{t+2}}(t)$. Thus $\hat{V}(\underline{b}_t, t) < \hat{V}(\underline{c}_t, t)$, which contradicts the definition of $\underline{b}_t$.

Hence, we just proved by contradiction that there exists $\ell \in \{L_{t+1}, L_{t+2}\}$ such that $\gamma \in \mathcal{I}_\ell(t)$. To prove Lemma 14, it remains to establish the following three statements.

1. $(\gamma \in \mathcal{I}_{L_{t+1}}(t)) \cap (\gamma \in \mathcal{I}_{L_{t+2}}(t)) \implies \left(\exists \ell \in \{L_{t+1}, L_{t+2}\} : \sqrt{\frac{2\beta(t,\delta)}{N_\ell(t)}} > \epsilon\right)$

2. $(\gamma \in \mathcal{I}_{L_{t+1}}(t)) \cap (\gamma \notin \mathcal{I}_{L_{t+2}}(t)) \implies \left(\sqrt{\frac{2\beta(t,\delta)}{N_{L_{t+1}}(t)}} > \epsilon\right)$

3. $(\gamma \notin \mathcal{I}_{L_{t+1}}(t)) \cap (\gamma \in \mathcal{I}_{L_{t+2}}(t)) \implies \left(\sqrt{\frac{2\beta(t,\delta)}{N_{L_{t+2}}(t)}} > \epsilon\right)$

*Statement 1.* As the algorithm doesn't stop, $U_{\underline{c}_t}(t) - L_{\underline{b}_t}(t) > \epsilon$. Hence

$$U_{L_{t+2}}(t) - L_{L_{t+1}}(t) \;>\; \epsilon$$

$$\hat{\mu}_{L_{t+2}}(t) + \sqrt{\frac{\beta(N_{L_{t+2}}(t),\delta)}{2N_{L_{t+2}}(t)}} - \hat{\mu}_{L_{t+1}}(t) + \sqrt{\frac{\beta(N_{L_{t+1}}(t),\delta)}{2N_{L_{t+1}}(t)}} \;>\; \epsilon$$

$$\sqrt{\frac{\beta(N_{L_{t+2}}(t),\delta)}{2N_{L_{t+2}}(t)}} + \sqrt{\frac{\beta(N_{L_{t+1}}(t),\delta)}{2N_{L_{t+1}}(t)}} \;>\; \epsilon$$

using that by definition of $\underline{b}_t$, $\hat{\mu}_{L_{t+2}}(t) < \hat{\mu}_{L_{t+1}}(t)$. Hence, there exists $\ell \in \{L_{t+1}, L_{t+2}\}$ such that

$$\sqrt{\frac{\beta(N_\ell(t),\delta)}{2N_\ell(t)}} > \frac{\epsilon}{2} \;\Rightarrow\; \sqrt{\frac{2\beta(t,\delta)}{N_\ell(t)}} > \epsilon.$$

*Statement 2.* We consider two cases and first assume that $\gamma \in \mathcal{I}_{L_{t+1}}(t)$ and $\gamma \geq U_{L_{t+2}}(t)$. Using the fact that the algorithm doesn't stop at round $t$, the following events hold

$$\begin{aligned}
& (U_{L_{t+2}}(t) - L_{L_{t+1}}(t) > \epsilon) \cap (U_{L_{t+1}}(t) > \gamma) \cap (U_{L_{t+2}}(t) \leq \gamma) \\
\Rightarrow\; & (U_{L_{t+2}}(t) - L_{L_{t+1}}(t) > \epsilon) \cap (U_{L_{t+1}}(t) > \gamma) \cap (U_{L_{t+2}}(t) - L_{L_{t+1}}(t) + L_{L_{t+1}}(t) \leq \gamma) \\
\Rightarrow\; & (U_{L_{t+1}}(t) > \gamma) \cap (L_{L_{t+1}}(t) \leq \gamma - \epsilon) \\
\Rightarrow\; & (U_{L_{t+1}}(t) - L_{L_{t+1}}(t) > \epsilon) \\
\Rightarrow\; & \left(\sqrt{\frac{2\beta(t,\delta)}{N_{L_{t+1}}(t)}} > \epsilon\right).
\end{aligned}$$

The second case is $\gamma \in \mathcal{I}_{L_{t+1}}(t)$ and $\gamma \leq L_{L_{t+2}}(t)$. Then the following holds

$$\begin{aligned}
& (U_{L_{t+2}}(t) - L_{L_{t+1}}(t) > \epsilon) \cap (L_{L_{t+1}}(t) \leq \gamma) \cap (L_{L_{t+2}}(t) \geq \gamma) \\
\Rightarrow\; & (L_{L_{t+1}}(t) \leq \gamma) \cap (U_{L_{t+2}}(t) + L_{L_{t+2}}(t) - L_{L_{t+1}}(t) > \gamma + \epsilon) \\
\Rightarrow\; & (L_{L_{t+1}}(t) \leq \gamma) \cap \left(2\hat{\mu}_{L_{t+2}}(t) - \hat{\mu}_{L_{t+1}}(t) + \sqrt{\frac{2\beta(N_{L_{t+1}}(t),\delta)}{2N_{L_{t+1}}(t)}} > \gamma + \epsilon\right) \\
\Rightarrow\; & (L_{L_{t+1}}(t) \leq \gamma) \cap \left(\hat{\mu}_{L_{t+1}}(t) + \sqrt{\frac{2\beta(N_{L_{t+1}}(t),\delta)}{2N_{L_{t+1}}(t)}} > \gamma + \epsilon\right) \\
\Rightarrow\; & (U_{L_{t+1}}(t) - L_{L_{t+1}}(t) > \epsilon),
\end{aligned}$$

where the third implication uses the fact that $\hat{\mu}_{L_{t+2}}(t) \leq \hat{\mu}_{L_{t+1}}(t)$.

*Statement 3.* We consider two cases and first assume that $\gamma \in \mathcal{I}_{L_{t+2}}(t)$ and $\gamma \leq L_{L_{t+1}}(t)$. Using the fact that the algorithm doesn't stop at round $t$, the following events hold

$$\begin{aligned}
& (U_{L_{t+2}}(t) - L_{L_{t+1}}(t) > \epsilon) \cap (L_{L_{t+1}}(t) \geq \gamma) \cap (L_{L_{t+2}}(t) \leq \gamma) \\
\Rightarrow\; & (U_{L_{t+2}}(t) > \gamma + \epsilon) \cap (L_{L_{t+2}}(t) \leq \gamma) \\
\Rightarrow\; & (U_{L_{t+2}}(t) - L_{L_{t+2}}(t) > \epsilon) \\
\Rightarrow\; & \left(\sqrt{\frac{2\beta(t,\delta)}{N_{L_{t+2}}(t)}} > \epsilon\right).
\end{aligned}$$

The second case is $\gamma \in \mathcal{I}_{L_{t+2}}(t)$ and $\gamma \geq \mathrm{U}_{L_{t+1}}(t)$. Then the following holds

$$
\begin{aligned}
&\left(\mathrm{U}_{L_{t+2}}(t) - \mathrm{L}_{L_{t+1}}(t) > \epsilon\right) \cap \left(\mathrm{U}_{L_{t+2}}(t) \geq \gamma\right) \cap \left(\gamma \geq \mathrm{U}_{L_{t+1}}(t)\right) \\
\Rightarrow \quad &\left(\mathrm{U}_{L_{t+2}}(t) - \left(\mathrm{L}_{L_{t+1}}(t) + \mathrm{U}_{L_{t+1}}(t)\right) > \epsilon - \gamma\right) \cap \left(\mathrm{U}_{L_{t+2}}(t) \geq \gamma\right) \\
\Rightarrow \quad &\left(\hat{\mu}_{L_{t+2}}(t) + \sqrt{\frac{2\beta(N_{L_{t+2}}(t),\delta)}{2N_{L_{t+2}}(t)}} - 2\hat{\mu}_{L_{t+1}}(t) > \epsilon - \gamma\right) \cap \left(\mathrm{U}_{L_{t+2}}(t) \geq \gamma\right) \\
\Rightarrow \quad &\left(-\hat{\mu}_{L_{t+2}}(t) + \sqrt{\frac{2\beta(N_{L_{t+2}}(t),\delta)}{2N_{L_{t+2}}(t)}} > \epsilon - \gamma\right) \cap \left(\mathrm{U}_{L_{t+2}}(t) \geq \gamma\right) \\
\Rightarrow \quad &\left(\mathrm{U}_{L_{t+2}}(t) - \mathrm{L}_{L_{t+2}}(t) > \epsilon\right),
\end{aligned}
$$

where the third implication uses the fact that $\hat{\mu}_{L_{t+2}}(t) \leq \hat{\mu}_{L_{t+1}}(t)$.

# E  Proof of the lower bounds

## E.1  Proof of Theorem 7

Theorem 7 follows from considering the best possible change of distribution $\boldsymbol{\lambda} \in \mathrm{Alt}(\boldsymbol{\mu})$. The expected log-likelihood ratio of the observations until $\tau$ under a model parameterized by $\boldsymbol{\mu}$ and a model parameterized by $\boldsymbol{\lambda}$ is

$$
\mathbb{E}_{\boldsymbol{\mu}}[L_\tau(\boldsymbol{\mu},\boldsymbol{\lambda})] = \sum_{\ell \in \mathcal{L}} \mathbb{E}_{\boldsymbol{\mu}}[N_\ell(\tau)] d(\mu_\ell, \lambda_\ell),
$$

where $N_\ell(t)$ is the number of draws of the leaf $\ell$ until round $t$. Using Lemma 1 of [15], for any event $\mathcal{E}$ in the filtration generated by $\tau$,

$$
\mathbb{E}_{\boldsymbol{\mu}}[L_\tau(\boldsymbol{\mu},\boldsymbol{\lambda})] \geq d(\mathbb{P}_{\boldsymbol{\mu}}(\mathcal{E}), \mathbb{P}_{\boldsymbol{\lambda}}(\mathcal{E})).
$$

As the strategy is $\delta$-correct, letting $\mathcal{E} = (\hat{s}_\tau = s^*(\boldsymbol{\mu}))$ one has $\mathbb{P}_{\boldsymbol{\mu}}(\mathcal{E}) \geq 1 - \delta$ and $\mathbb{P}_{\boldsymbol{\lambda}}(\mathcal{E}) \leq \delta$ (under this model, $s^*(\boldsymbol{\mu})$ is not the best action at the root under the model parameterized by $\boldsymbol{\lambda}$). Using monotonicity properties of the Bernoulli KL-divergence, one obtains, for any $\boldsymbol{\lambda} \in \mathrm{Alt}(\boldsymbol{\mu})$,

$$
\sum_{\ell \in \mathcal{L}} \mathbb{E}_{\boldsymbol{\mu}}[N_\ell(\tau)] d(\mu_\ell, \lambda_\ell) \geq d(1 - \delta, \delta).
$$

Then, one can write

$$
\begin{aligned}
\inf_{\boldsymbol{\lambda} \in \mathrm{Alt}(\boldsymbol{\mu})} \sum_{\ell \in \mathcal{L}} \mathbb{E}_{\boldsymbol{\mu}}[N_\ell(\tau)] d(\mu_\ell, \lambda_\ell) &\geq d(1 - \delta, \delta) \\
\mathbb{E}_{\boldsymbol{\mu}}[\tau] \inf_{\boldsymbol{\lambda} \in \mathrm{Alt}(\boldsymbol{\mu})} \sum_{\ell \in \mathcal{L}} \frac{\mathbb{E}_{\boldsymbol{\mu}}[N_\ell(\tau)]}{\mathbb{E}_{\boldsymbol{\mu}}[\tau]} d(\mu_\ell, \lambda_\ell) &\geq d(1 - \delta, \delta) \\
\mathbb{E}_{\boldsymbol{\mu}}[\tau] \left(\sup_{\boldsymbol{w} \in \Sigma_{|\mathcal{L}|}} \inf_{\boldsymbol{\lambda} \in \mathrm{Alt}(\boldsymbol{\mu})} \sum_{\ell \in \mathcal{L}} \frac{\mathbb{E}_{\boldsymbol{\mu}}[N_\ell(\tau)]}{\mathbb{E}_{\boldsymbol{\mu}}[\tau]} d(\mu_\ell, \lambda_\ell)\right) &\geq d(1 - \delta, \delta),
\end{aligned}
$$

using that $\sum_{\ell \in \mathcal{L}} \frac{\mathbb{E}_{\boldsymbol{\mu}}[N_\ell(\tau)]}{\mathbb{E}_{\boldsymbol{\mu}}[\tau]} = 1$. This concludes the proof.

One can also note that for an algorithm to match the lower bound, all the inequalities above should be equalities. In particular one would need $w_\ell^*(\boldsymbol{\mu}) \simeq \frac{\mathbb{E}_{\boldsymbol{\mu}}[N_\ell(\tau)]}{\mathbb{E}_{\boldsymbol{\mu}}[\tau]}$, where $w_\ell^*(\boldsymbol{\mu})$ is a maximizer in the definition of $T^*(\boldsymbol{\mu})^{-1}$ in (4).

## E.2  Proof of Lemma 8

In the particular case of a depth-two tree with $K$ actions for player A and $M$ actions for player B,

$$
T^*(\boldsymbol{\mu})^{-1} = \sup_{\boldsymbol{w} \in \Sigma_{K \times M}} \inf_{\boldsymbol{\lambda} \in \mathrm{Alt}(\boldsymbol{\mu})} \sum_{k=1}^{K} \sum_{m=1}^{K} w_{k,m} d(\mu_{k,m}, \lambda_{k,m}).
$$

From the particular structure of $\boldsymbol{\mu}$, the best action at the root is action $i = 1$. Hence

$$
\mathrm{Alt}(\boldsymbol{\mu}) = \{\boldsymbol{\lambda} : \exists a \in \{1, \ldots, M\}, \exists i \in \{2, \ldots, K\} : \forall j \in \{1, \ldots, M\}, \lambda_{1,a} < \lambda_{i,j}\}.
$$

It follows that

$$
\begin{aligned}
T^*(\boldsymbol{\mu})^{-1} &= \sup_{\boldsymbol{w}\in\Sigma_{K\times M}} \min_{\substack{a\in\{1,\ldots,M\}\\ i\in\{2,\ldots,K\}}} \inf_{\boldsymbol{\lambda}:\forall j,\lambda_{1,a}<\lambda_{i,j}} \sum_{k=1}^{K}\sum_{m=1}^{M} w_{k,m}d(\mu_{k,m},\lambda_{k,m}) \\
&= \sup_{\boldsymbol{w}\in\Sigma_{K\times M}} \min_{\substack{a\in\{1,\ldots,M\}\\ i\in\{2,\ldots,K\}}} \underbrace{\inf_{\boldsymbol{\lambda}:\forall j,\lambda_{1,a}<\lambda_{i,j}} \left[ w_{1,a}d(\mu_{1,a},\lambda_{1,a}) + \sum_{j=1}^{M} w_{i,j}d(\mu_{i,j},\lambda_{i,j}) \right]}_{:=F_{a,i}(\boldsymbol{\mu},\boldsymbol{w})}.\quad(15)
\end{aligned}
$$

Indeed, in the rightmost constrained minimization problem, all the $\lambda_{k,m}$ on which no constraint lie can be set to $\mu_{k,m}$ to minimize the corresponding term in the sum. Using tools from constrained optimization, one can prove that

$$
F_{a,i}(\boldsymbol{\mu},\boldsymbol{w}) = \inf_{(\lambda_{1,a},(\lambda_{i,j})_j)\in\mathcal{C}} \left[ w_{1,a}d(\mu_{1,a},\lambda_{1,a}) + \sum_{j=1}^{M} w_{i,j}d(\mu_{i,j},\lambda_{i,j}) \right],
$$

where $\mathcal{C}$ is the set

$$
\mathcal{C} = \{(\mu',\boldsymbol{\mu}')\in[0,1]^{M+1} : \exists j_0 \in\{1,\ldots,K\}, \exists c \in [\mu_{i,j_0},\mu_{1,a}] : \mu' = \boldsymbol{\mu}'_1 = \cdots = \boldsymbol{\mu}'_{j_0} = c \text{ and } \boldsymbol{\mu}'_j = \mu_{i,j} \text{ for } j > j_0\}.
$$

Letting $H(\mu',\boldsymbol{\mu}',w,\boldsymbol{w}) = w_{1,a}d(\mu_{1,a},\mu') + \sum_{j=1}^{M} w_{i,j}d(\mu_{i,j},\boldsymbol{\mu}'_j)$ one can easily show that for all $(\mu',\boldsymbol{\mu}')\in\mathcal{C}$,

$$
H(\mu',\boldsymbol{\mu}',w,\boldsymbol{w}) \leq H(\mu',\boldsymbol{\mu}',w,\tilde{\boldsymbol{w}}),
$$

where $\tilde{\boldsymbol{w}}$ is constructed from $\boldsymbol{w}$ by putting all the weight on the smallest arm:

$$
\tilde{w}_{i,1} = \sum_{j\geq 1} w_{1,j} \text{ and } \tilde{w}_{i,j} = 0 \text{ for } j \geq 2.
$$

This is because the largest $d(\mu_{i,j},c)$ is $d(\mu_{i,1},c)$ as $\mu_{i,1} \leq \mu_{i,j} \leq c$ for $j \leq j_0$. Hence taking the infimum, one obtains

$$
F_{a,i}(\boldsymbol{\mu},\boldsymbol{w}) \leq F_{a,i}(\boldsymbol{\mu},\tilde{\boldsymbol{w}}).
$$

Repeating this argument for all $i$, one can construct $\tilde{\boldsymbol{w}}$ such that

$$
\forall i \geq 2, \tilde{w}_{i,1} = \sum_{j\geq 1} w_{i,j} \text{ and } \tilde{w}_{i,j} = 0 \text{ for } j \geq 2
$$

and $F_{a,i}(\boldsymbol{\mu},\boldsymbol{w}) \leq F_{a,i}(\boldsymbol{\mu},\tilde{\boldsymbol{w}})$ for all $a,i$. Thus, the supremum in 15 is necessarily attained for $\boldsymbol{w}$ in the set $\tilde{\Sigma}_{K\times M} = \{\boldsymbol{w}\in\Sigma_{K,M} : w_{i,j} = 0 \text{ for } i \geq 2, j \geq 2\}$. It follows that

$$
\begin{aligned}
T^*(\boldsymbol{\mu})^{-1} &= \sup_{\boldsymbol{w}\in\tilde{\Sigma}_{M+K-1}} \min_{\substack{a\in\{1,\ldots,M\}\\ i\in\{2,\ldots,K\}}} \inf_{\boldsymbol{\lambda}:\forall j,\lambda_{1,a}<\lambda_{i,1}} \left[ w_{1,a}d(\mu_{1,a},\lambda_{1,a}) + w_{i,1}d(\mu_{i,1},\lambda_{i,1}) \right] \\
&= \sup_{\boldsymbol{w}\in\tilde{\Sigma}_{K+M-1}} \min_{\substack{a=1,\ldots,M\\ i=2,\ldots,K}} \left[ w_{1,a}d\left(\mu_{1,a},\frac{w_{1,a}\mu_{1,a}+w_{i,1}\mu_{i,1}}{w_{1,a}+w_{i,1}}\right) + w_{i,1}d\left(\mu_{i,1},\frac{w_{1,a}\mu_{1,a}+w_{i,1}\mu_{i,1}}{w_{1,a}+w_{i,1}}\right) \right],
\end{aligned}
$$

which concludes the proof.

## F    Inverting Bounds

Consider the exploration rate

$$
\beta(s) = C + \frac{3}{2}\ln(1+\ln s) \qquad \text{where} \qquad C = \ln\frac{|\mathcal{L}|}{\delta} + 3\ln\ln\frac{|\mathcal{L}|}{\delta}
$$

where we assume $C \geq 1$, so that $\beta(1) = C \geq 1$. Now fix some $a \geq 1$, and let us define

$$
S = \sup\{s \geq 1 | a\beta(s) \geq s\}.
$$

The goal is to get a tight upper bound on $S$. We claim that

**Theorem 16.**

$$aC + \frac{3}{2}a\ln(1+\ln(aC)) \leq S \leq aC + \frac{3}{2}a\ln(1+\ln(aC))\underbrace{\frac{C(1+\ln(aC))}{C(1+\ln(aC)) - \frac{3}{2}}}_{\to 1},$$

*where the upper bound is only non-trivial if $C(1+\ln(aC)) > \frac{3}{2}$, which, for example, is implied by $C > 1.23696$.*

*Proof.* The requirement $a\beta(s) \geq s$ is monotone in $s$, in that it holds for small $s$ (including $s = 1$) and fails for large $s$. So to show the theorem it suffices to show that $a\beta(s) \geq s$ holds at $s$ given by the left-hand-side, while it fails for $s$ equal to the right-hand side.

First, we need to establish that $a\beta(s) \geq s$ for $s$ equal to the left-hand side expression of the theorem. That is, we need to show

$$a\left(C + \frac{3}{2}\ln\left(1 + \ln\left(a\left(C + \frac{3}{2}\ln(1 + \ln(aC))\right)\right)\right)\right) \geq a\left(C + \frac{3}{2}\ln(1 + \ln(aC))\right)$$

which is equivalent (the simplification is entirely mechanical) to

$$aC \geq 1$$

which holds by assumption. Then we need to plug in the right hand side. Here we need to show that

$$a\left(C + \frac{3}{2}\ln\left(1 + \ln\left(aC + \frac{3}{2}a\ln(1 + \ln(aC))\frac{C(1+\ln(aC))}{C(1+\ln(aC)) - \frac{3}{2}}\right)\right)\right)$$
$$\leq aC + \frac{3}{2}a\ln(1 + \ln(aC))\frac{C(1+\ln(aC))}{C(1+\ln(aC)) - \frac{3}{2}},$$

that is

$$\ln\left(1 + \ln\left(aC + \frac{3}{2}a\ln(1 + \ln(aC))\frac{C(1+\ln(aC))}{C(1+\ln(aC)) - \frac{3}{2}}\right)\right) \leq \ln(1 + \ln(aC))\frac{C(1+\ln(aC))}{C(1+\ln(aC)) - \frac{3}{2}}.$$

To show this, we use $\ln(1+x) \leq x$ twice to show

$$\ln\left(1 + \ln\left(aC + \frac{3}{2}a\ln(1 + \ln(aC))\frac{C(1+\ln(aC))}{C(1+\ln(aC)) - \frac{3}{2}}\right)\right)$$

$$= \ln\left(1 + \ln(aC) + \ln\left(1 + \frac{3}{2}\ln(1 + \ln(aC))\frac{(1+\ln(aC))}{C(1+\ln(aC)) - \frac{3}{2}}\right)\right)$$

$$\leq \ln\left(1 + \ln(aC) + \frac{3}{2}\ln(1 + \ln(aC))\frac{(1+\ln(aC))}{C(1+\ln(aC)) - \frac{3}{2}}\right)$$

$$= \ln(1 + \ln(aC)) + \ln\left(1 + \frac{\frac{3}{2}\ln(1+\ln(aC))}{C(1+\ln(aC)) - \frac{3}{2}}\right)$$

$$\leq \ln(1 + \ln(aC)) + \frac{\frac{3}{2}\ln(1+\ln(aC))}{C(1+\ln(aC)) - \frac{3}{2}}$$

$$= \frac{C(1+\ln(aC))}{C(1+\ln(aC)) - \frac{3}{2}}\ln(1 + \ln(aC))$$

as desired. □