[Reviews · NeurIPS 2017]

Reviewer 1



This work uses best arm identification (BAI) techniques applies to the monte-carlo tree search problem with two-players (in a turn-based setting). The goal is to find the best action for player A to take by carefully considering all the next actions that player B and A can take in the following rounds. Access to a stochastic oracle to evaluate the values of leaves is supposed, hence the goal is to find approximately (with precision \epsilon) the best action at the root with high confidence (at least 1-\delta). Algorithms based on confidence intervals and upwards propagation (from leaf to the root) of the upper (for the MAX nodes, action by player A) and lower (for the MIN nodes, action by player B the opponent) confidence bounds are proposed. The algorithms are intuitive and well described, and well rooted in the fixed confidence BAI literature. In particular, recent refinements in this line of work are used, but also well separated from the new contributions (Section 3.1) which makes the paper very clear. A theorem on the number of leaf evaluations is given, and the complexity H (3) exhibited and improves on previous work. The comparison with existing algorithms and strategies is precise. Finally, experiments show a big improvement over existing methods, and some insights are given on lower bounds and future directions, which again are grounded in recent results in the BAI fixed confidence setting. The paper is very well written, and the contributions are made easy to understand and contextualize, thank you. Some comments to maybe further improve the clarity of the work. Figure 1 should be pointed to in the text (it is easier to understand immediately what’s going on than line 111). There could be a discussion to explain why you need two candidates a_t and b_t at time t+1, which can be surprising at first. In the experiments, why do you set \delta = 0.1 |\mathcal L|? It is strange in Figure 4 to have \delta = 2.7 (though experimentally this of course does not seem to be a problem). Is it common in practice to run these algorithms with such little confidence parameter? I would suggest running the experiments with \delta = 0.01*27 instead to avoid confusion, or just \delta = 0.1 (without L). Lemma 7 is hard to parse in its current form, but the example makes this task easier. Could it be possible to add a line in Figure 3 with the optimal proportions (making sure the same \delta is used)? (extremely minor but 259.9 x 1.76 = 457.424, maybe remove the numerical value of kl(\delta, 1-\delta)) Figure 4 is unreadable without 300% zoom on my computer, maybe it could be removed. I suggest the following change to the figures: adding two columns with name of the algorithm and the total number of samples used, and then displaying proportions instead of raw numbers (same as w^* ine line 281). Overall, I believe this is a clear and significant contribution with respect to the existing literature.

Reviewer 2



This paper treats the problem of Monte Carlo Tree Search in depth-k game trees. Even though the problem was considered last year for depth-two trees in 'Optimal best arm identification with fixed confidence' (COLT 2016), the present paper considerably extends this previous work that could not directly be generalized to higher depths. They propose a new type of algorithm that can be run with different Best Arm Identification (BAI) algorithms and they provide the analysis for LUCB and UGapE. The idea relies on two main ingredients : First, the BAI algorithm leads the learner in its first decision departing from the root of the tree and then, the subsequent 'representative leaf' is sampled. This leaf is chosen by following a somehow greedy path along the tree, each time choosing the action that maximizes (resp. minimizes) the expected value. This line of work sheds a new light on MCTS problems by bringing the tools developed in the framework of Best Arm Identification. The proposed method has several qualities : 1. It is fairly intuitive and easy to understand for the reader having some ground knowledge of the BAI literature; 2. It is computationally efficient, allowing to conduct convincing experiments; 3. It outperforms the existing algorithm FindTopWinner (see comment below). The authors also prove a problem-dependent lower bound following the now classic technics from e.g. (Garivier and Kaufmann, 2016), (Kaufmann et al., 2015). Even though the final result of Lemma 7 is not fully explicit, the optimization problem can be numerically solved. This allows in particular to have a comparison basis for the experimental section, on top of existing algorithms. Also, as mentioned in the comments of Lemma 7, such a lower bound result gives a hope towards finally obtaining an asymptotically optimal algorithm following the Track-And-Stop technique from (Garivier and Kaufmann, 2016). This is indeed an exciting perspective, coming with a few other new problems opened by this paper. I only have minor comments: - The Lemmas 13 and 14 seems to be the key tools that allow to control the behavior of the algorithm with respect to the choice of the 'representative leaf'. However, this intuition is not really clear in the proof of Theorem 3 that is sketched in the main paper. Space requirements don't make it easy but this could make the idea of the proof even clearer. - The 'update information about the arms' in the Figure 2 is quite unclear. Actually, the exact action of the algorithm is a bit tricky, I had to read several times to understand whether the algorithm only chooses the first step or all the path of actions and where the information/feedback finally shows up. This could be made a bit clearer in the algorithm pseudo-code maybe. - l 255 : 'uniformLY at random'; - Lemma 11 in Appendix, please chose i or k to index s; Overall, I strongly recommend accepting this paper as it opens a new path towards the understanding of MCTS problems and may lead to future major advances.

Reviewer 3



\noindent The paper considers the game tree of a two-person zero-sum game. The payoff at each leaf of the tree is assumed to be stochastic and the tree can be of arbitrary length. The goal is to identify the best move at the root of the tree that gives the highest payoff. A PAC setting is considered in this paper where the move can be sub-optimal within a parameter $\epsilon$. Notice that the Best Arm Identification (MAI) problem is such a tree with depth 1. Two algorithms are proposed which are based respectively on UGapE and LUCB. They are essentially the same as in the MAI problem. However, in the present context, the algorithms need to deal with the structure (i.e. depth, branches) of the tree. The algorithms maintain confidence intervals at each leaf node and propagate them upwards to the root. They also maintain a representative" leaf for each internal node of the tree, which corresponds to the leaf that should be sampled if the confidence interval of that internal node needs to be updated. Then the algorithm proceeds similarly as in MAI. In each round, the algorithm identifies the two marginal children of the root, and samples the represetative leaf of one (or both, depending on the algorithm) of them. The stopping rule is exactly the same as the one typically used for MAI. Theoretical upper and lower bounds are derived. Upper bounds are similar to the results of LUCB in MAI and lower bounds are also similar to that in MAI. In general, the paper is well-motivated and the results presented in the current paper indeed improves upon earlier results both theoretically and experimentally (especially experimentally, in which the new algorithms take only about 1/7 the number of samples by the state-of-the-art algorithm). Moreover, the paper is clear written and easy to follow. However, while the problem considered is a generalization of the MAI problem, the algorithms and all the analysis are too similar to that in MAI. In addition, while the paper by Jamieson et al. 2014 on LIL is listed in the reference, it is not mentioned if that techinique will help to improve the results both experimentally and theoretically. It it known that LIL can improve the CBlike bound in MAI setting. It would be better to investigate this point. To sum up, the paper is in general a good paper, especially given the significant improvement in experimental results.